# Metformin and Everolimus: A Promising Combination for Neuroendocrine Tumors Treatment

**DOI:** 10.3390/cancers12082143

**Published:** 2020-08-02

**Authors:** Eleonora Vitali, Ilena Boemi, Giulia Tarantola, Sara Piccini, Alessandro Zerbi, Giulia Veronesi, Roberto Baldelli, Gherardo Mazziotti, Valeria Smiroldo, Elisabetta Lavezzi, Anna Spada, Giovanna Mantovani, Andrea G. Lania

**Affiliations:** 1Laboratory of Cellular and Molecular Endocrinology, Humanitas Clinical and Research Center—IRCCS (Istituto di Ricovero e Cura a Carattere Scientifico), 20089 Rozzano, Italy; ilena.boemi@humanitasresearch.it (I.B.); Giulia.tarantola23@gmail.com (G.T.); sara.piccini@humanitas.it (S.P.); 2Department of Biomedical Sciences, Humanitas University, 20090 Pieve Emanuele, Italy; alessandro.zerbi@hunimed.eu (A.Z.); gherardo.mazziotti@hunimed.eu (G.M.); andrea.lania@hunimed.eu (A.G.L.); 3Pancreas Surgery Unit, Humanitas Clinical and Research Center—IRCCS, 20089 Rozzano, Italy; 4School of Medicine, Vita-Salute San Raffaele University, 20100 Milan, Italy; veronesi.giulia@hsr.it; 5Division of Thoracic Surgery, IRCCS San Raffaele Scientific Institute, 20100 Milan, Italy; 6Endocrinological Oncology, Service of Endocrinology, A.O. San Camillo-Forlanini, 13449 Rome, Italy; RBaldelli@scamilloforlanini.rm.it; 7Endocrinology, Diabetology and Andrology Unit, Humanitas Clinical and Research Center—IRCCS, 20089 Rozzano, Italy; elisabetta.lavezzi@humanitas.it; 8Oncology Unit, Humanitas Clinical and Research Center—IRCCS, 20089 Rozzano, Italy; valeria.smiroldo@cancercenter.humanitas.it; 9Department of Clinical Sciences and Community Health, University of Milan, 20100 Milan, Italy; anna.spada@unimi.it (A.S.); giovanna.mantovani@unimi.it (G.M.); 10Endocrinology Unit, Fondazione IRCCS Ca’ Granda Ospedale Maggiore Policlinico, 20100 Milan, Italy

**Keywords:** metformin, everolimus, neuroendocrine tumors, resistance

## Abstract

Introduction: Treatment options for neuroendocrine tumors (NETs) are rarely curative, as NETs frequently show resistance to medical therapy. The use of everolimus, an mTOR inhibitor, is limited by the development of resistance, probably due to the activation of Akt signaling. In this context, the antidiabetic drug metformin is able to inhibit mTOR, providing a rationale for the use of metformin and everolimus in combination. Methods: We investigated the effects of the metformin and everolimus combination on NET cell proliferation, apoptosis, colony formation, cell viability, NET spheroids growth and the involvement of the Akt and mTOR pathways, and also developed everolimus-resistant NET cells to further study this combination. Results: Metformin and everolimus in combination are more effective than monotherapy in inhibiting pancreatic NET (PAN-NET) cell proliferation (−71% ± 13%, *p* < 0.0001 vs. basal), whereas no additive effects were observed on pulmonary neuroendocrine tumor (PNT) cell proliferation. The combinatorial treatment is more effective than monotherapy in inhibiting colony formation, cell viability, NET spheroids growth rate and mTOR phosphorylation in both NET cell lines. In a PAN-NET cell line, metformin did not affect Akt phosphorylation; conversely, it significantly decreased Akt phosphorylation in a PNT cell line. Using everolimus-resistant NET cells, we confirmed that metformin maintained its effects, acting by two different pathways: Akt-dependent or independent, depending on the cell type, with both leading to mTOR suppression. Conclusions: Considering the promising effects of the everolimus and metformin combination in NET cells, our results provide a rationale for its use in NET patients.

## 1. Introduction

Neuroendocrine tumors (NETs) are considered a rare malignancy. However, their incidence has rapidly increased in the last decades. NETs are more frequently localized in the gastrointestinal tract (60–70%) and in the lungs (20–30%). The only treatment that offers a cure is surgery. However, up to 50% of patients are diagnosed with metastatic disease, and curative surgery is usually not an option [1,2]. These patients can be offered long-term systemic treatment, for both symptomatic relief and tumor growth suppression. Treatment options include somatostatin analogs, the mTOR inhibitor everolimus, the tyrosine kinase inhibitor sunitinib and peptide receptor radionuclide therapy, alone or combined with cytoreductive procedures. Unfortunately, a significant number of NET patients do not respond to the above-mentioned medical treatments and the molecular mechanisms involved are still not clearly understood [3,4,5].

The demonstration of the important role of the PI3K-Akt-mTOR pathway in pancreatic NET (PAN-NET) tumorigenesis [6] and the finding of an abnormal activation of mTOR—a serine/threonine protein kinase—in patients with PAN-NET [7] led to the use of the mTOR inhibitor everolimus for the treatment of pulmonary neuroendocrine tumors (PNTs) and PAN-NETs [8,9]. Unfortunately, the use of everolimus is limited by the development of resistance, which has been reported both in vivo and in vitro [9,10]. The mechanisms leading to resistance are not well understood, but activation of a negative feedback leading to the activation of Akt and the consequent mTOR reactivation has been proposed as a possible resistance mechanism. In particular, mTOR inhibition induced insulin receptor substrate 1 (IRS-1) expression and abrogated feedback inhibition of the pathway, leading to Akt activation in cancer cell lines and in primary cell cultures [11]. Indeed, hyperactivation of PI3K/Akt may contribute to PAN-NET development and behavior [6].

Combination therapy to ablate mTOR function and prevent Akt activation may improve antitumor activity. Indeed, the combination of everolimus and a PIK3 inhibitor (LY29400) inhibited the overexpression of pAkt, decreased cell growth in PAN-NET cell lines (BON cell line) and progression of liver metastasis in vivo [12]. However, Akt/PI3K inhibitors may have a limited clinical role because of unacceptable toxicity in patients [13,14].

Interestingly, the use of metformin (1.1-dimethylbiguanide hydrochloride), a widely used antidiabetic drug, was associated with significantly lower risks of cancer incidence and mortality [15]. It displays potent anticancer properties in different types of cancers [16,17,18,19], being able to suppress mammalian target of rapamycin (mTOR).

Although the most well-known mechanism of metformin action is adenosine monophosphate-activated protein kinase (AMPK) activation, recent investigations have shown that AMPK-independent pathways can explain some of metformin’s beneficial metabolic effects (i.e., suppression of glucagon signaling, activation of autophagy, attenuation of inflammasome activation) [20]. In this respect, a significant association between metformin use and longer PFS (progression-free survival) in patients affected by advanced PAN-NETs [21] has been found. Recently, we demonstrated the efficacy of metformin in inhibiting pancreatic neuroendocrine tumor cell line (QGP-1) proliferation in a dose- and time-dependent manner. Moreover, metformin exerted its anticancer effects, inhibiting mTOR phosphorylation through AIP (aryl hydrocarbon receptor-interacting protein) modulation in QGP-1 cells [22].

Importantly, metformin stimulates phosphorylation of IRS-1, which transmits signals from the insulin and IGF-1 receptors to the PI3K–Akt pathway, leading to inhibition of IGF-1–insulin signaling. This activity itself has the potential to downregulate mTOR signaling [23]. Therefore, metformin, by inhibiting mTOR and blocking the IGFR-1/IRS-1/PI3K/Akt cascade, may have a role in overcoming everolimus resistance, suggesting a rationale for the use of metformin in combination with everolimus.

Although previous studies have demonstrated the anticancer properties of metformin, the impact of metformin in everolimus-resistant NET cells and the effects of metformin and everolimus in combination in NET cells are unknown.

The aims of this study are to evaluate the anticancer effects of the metformin and everolimus combination, the molecular mechanism involved in its action in pancreatic and pulmonary NET cells and the effect of metformin on everolimus-resistant NET cells.

## 2. Results

### 2.1. The Effect of Metformin and Everolimus on NET Cell Proliferation and Apoptosis

First, by performing cell proliferation assays, we evaluated the antiproliferative effects of metformin and everolimus, alone or in combination, on cultured cells obtained from surgically removed human NETs. As shown in Figure 1A,B, metformin and everolimus alone decreased cell proliferation in PAN-NETs (−44% ± 6%, *p* < 0.01 vs. basal and −35% ± 9%, *p* < 0.01 vs. basal, respectively) and PNTs (−70% ± 12%, *p* < 0.0001 vs. basal and −32% ± 3%, *p* < 0.01 vs. basal, respectively). Interestingly, in combination they were more effective than each monotherapy in inhibiting primary PAN-NET cell proliferation (−71% ± 13%, *p* < 0.0001 vs. basal, *p* < 0.05 vs. metformin and *p* < 0.01 vs. everolimus) (Figure 1A and Appendix A). On the contrary, the metformin and everolimus combination was not more effective than the single drugs on inhibiting cell proliferation in primary PNT cells (Figure 1B).

Thereafter, considering the low availability of NET tumor samples and the low yield in terms of viable cells from sample dispersion, to further elucidate the antiproliferative action of the metformin and everolimus combination, we used the QGP-1 and H727 cell lines as a model for PAN-NET s and PNTs, respectively.

As demonstrated in primary NETs, we confirmed that the metformin and everolimus combination was more effective than the monotherapies in inhibiting QGP-1 cell proliferation, with the greatest effect achieved using metformin at 10 mM and everolimus at 10 nM (−77% ± 13%, *p* < 0.0001 vs. basal, *p* < 0.01 vs. metformin and *p* < 0.05 vs. everolimus) (Figure 1C). Similarly, as shown in Figure 1D, both metformin and everolimus as monotherapies decreased H727 cell proliferation (−40% ± 5% at 10mM, *p* < 0.0001 vs. basal and −29% ± 7% at 10 nM, *p* < 0.001 vs. basal, respectively), but the combination was not more effective in inhibiting H727 cell growth.

To investigate the ability of metformin and everolimus to induce apoptosis, QGP-1 and H727 cells were incubated with either drug alone or in combination and analyzed by flow cytometry. As shown in Figure 1E,F, only metformin significantly increased QGP-1 and H727 cell apoptosis (233% ± 40%, *p* < 0.001 vs. basal; 192% ± 26%, *p* < 0.01 vs. basal; everolimus: 117% ± 36%, *p* = 0.85 vs. basal; 120% ± 6%, *p* = 0.62 vs. basal), while no additive effect was detected by everolimus co-incubation (187% ± 16%, *p* < 0.01 vs. basal; 157% ± 20%, *p* < 0.05 vs. basal).

### 2.2. The Effect of Metformin and Everolimus on QGP-1 and H727 Cell Viability

To investigate the antiproliferative effect of the metformin and everolimus combination after a longer incubation period, studies on cell viability were conducted in QGP-1 and H727 cells, using an MTT assay. Since the longer incubation time allowed the use of lower concentrations of metformin (100 μM or 400 μM), as recently demonstrated in QGP-1 cells [22], cells were treated with metformin (50 μM, 100 μM or 400 μM) and everolimus 10 nM, alone or in combination, and incubated for seven days. The incubation with either metformin (100 μM and 400 μM) or everolimus induced a significant reduction of cell viability in both cell lines (Figure 2). The combination of metformin 400 µM and everolimus 10 nM was significantly more effective than each monotherapy in reducing cell viability of both QGP-1 and H727 cells (−68% ± 9% and −61% ± 7%, *p* < 0.0001 vs. basal, respectively).

### 2.3. Metformin and Everolimus in Combination Are More Effective than Each Monotherapy in Inhibiting QGP-1 and H727 Colony Formation

To determine the role of the combination of metformin and everolimus in the ability of QGP-1 and H727 to grow in vitro and to form colonies, cells were incubated with either drug alone or with their combination for seven days. As shown in Figure 3A,B, in QGP-1 cells, metformin and everolimus alone significantly decreased the quantity of colonies, number of cells (−33% ± 13%, *p* < 0.01 metformin at 100 µM, −46% ± 13%, *p* < 0.001 metformin at 400 µM, −48% ± 22% everolimus at 10 nM, *p* < 0.0001) and colony size (Figure 3C) with respect to untreated cells. The combination of metformin 400 µM and everolimus 10 nM was more effective than each monotherapy (number of cells −73% ± 10%, *p* < 0.0001 vs. basal).

In H727 cell lines, metformin and everolimus alone significantly reduced the quantity of colonies, number of cells (−34% ± 15%, *p* < 0.001 metformin at 400 µM, −48% ± 7% everolimus at 10 nM, *p* < 0.0001) and colony size (Figure 3D–F) with respect to untreated cells. The combination of metformin 400 µM and everolimus 10 nM was more effective than each monotherapy (number of cells −70% ± 4% *p* < 0.0001), suggesting that the combination requires a longer incubation time in H727 cells to be effective in suppressing cell proliferation.

### 2.4. Metformin and Everolimus Treatment on QGP-1 and H727 3D Spheroids

Since the combination of metformin and everolimus has shown a promising additive antiproliferative effect in QGP-1 and H727 monolayers, we tested the effect of these compounds on the growth of QGP-1 and H727 3D spheroids. We observed the growth rate of spheroids after zero, three and six days of treatment (T0, T3, T6).

As shown in Figure 4, in untreated conditions, the size of QGP-1 and H727 spheroids increased in a time-dependent manner (T6: +59% ± 18% and +88% ± 25%, *p* < 0.05 and *p* < 0.0001 vs. T0, respectively). After six days of treatment with metformin or everolimus alone, in both QGP-1 and H727 3D models, the growth rate was significantly blocked with respect to the untreated control (in QGP-1: −17% ± 7% and −19% ± 1%, *p* < 0.05 and *p* < 0,01 vs. T6 basal; in H727: −20% ± 4% and −21% ± 1%, *p* < 0,0001 vs. T6 basal, respectively). Interestingly, the combination of metformin and everolimus induced a more potent effect on the growth rate of QGP-1 and H727 spheroids than monotherapy (−33% ± 12% and −35% ± 16%, *p* < 0,01 and *p* < 0,0001 vs. T6 basal, respectively).

### 2.5. The Effect of Metformin and Everolimus on mTOR and Akt Phosphorylation in NET Cells

Considering that everolimus is a mTOR inhibitor and metformin may be able to block the IGFR-1/IRS-1/PI3K/Akt/mTOR cascade, we analyzed the effects of the combination of the two drugs on the phosphorylation of mTOR to better understand the mechanism by which it exerts its anticancer activity.

As expected, metformin and everolimus alone decreased the levels of p-mTOR in both QGP-1 and H727 cell lines. Compared with each monotherapy, a significant reduction of p-mTOR levels was observed with the combination treatment (Figure 5A,B). These data suggest that the combination of metformin and everolimus induces a more potent mTOR blockade in both NET cell lines.

In order to investigate the mechanism responsible for the anticancer activity of the metformin and everolimus combination in NET cells, the effect of the two drugs on the phosphorylation of Akt was assessed by Western blot analysis. As shown in Figure 5C, metformin and everolimus alone or in combination did not exert any significant effect on Akt phosphorylation in QGP-1 cells. On the contrary, metformin significantly decreased the levels of p-Akt in H727 cell lines (Figure 5D), suggesting Akt involvement in metformin action in H727 cells.

### 2.6. The Effects of Metformin In Everolimus-Resistant QGP-1-R and H727-R

In order to study the molecular mechanism involved in everolimus resistance and to assess a novel therapeutic strategy, we induced resistance to everolimus in QGP-1 and H727 cell lines. After the establishment of the long term everolimus resistance, QGP-1-R and H727-R were treated with metformin and everolimus, alone or in combination, to perform cell proliferation assays. As expected, everolimus did not decrease QGP-1-R and H727-R cell proliferation, whereas metformin (10 mM) inhibited QGP-1-R and H727-R cell proliferation (−32% ± 10%, *p* < 0.0001 vs. basal, −36% ± 10%, *p* < 0.001 vs. basal, respectively) and this effect was maintained when the cells were incubated with the combination of metformin (10 mM) and everolimus (10 nM) (−31% ± 8%, *p* < 0.001 vs. basal, −25% ± 8%, *p* < 0.05 vs. basal, respectively), as shown in Figure 6A,B. To confirm these results, QGP-1-R and H727-R cells were incubated with either drug alone or with their combination and their effects on cell cycle were analyzed by flow cytometry. As shown in Figure 6C,D, metformin (10 mM) inhibited the growth of QGP-1-R and H727-R cells by blocking cell cycle progression at the G0/G1 phase and this effect was not affected when metformin and everolimus were used in combination. Interestingly, metformin kept promoting apoptosis in everolimus-resistant NET cells, alone (+155% ± 21%, *p* < 0.001 vs. basal in QGP-1-R and +117% ± 11%, *p* < 0.001 vs. basal in H727-R) or in combination with everolimus (+105% ± 13%, *p* < 0.01 vs. basal in QGP-1-R and +129% ± 10%, *p* < 0.001 vs. basal in H727-R) (Figure 6E,F).

Moreover, in order to investigate the pathway activated by metformin in everolimus-resistant NET cells, we tested the effects of metformin and everolimus on mTOR phosphorylation. As expected, everolimus did not decrease mTOR phosphorylation in QGP-1-R and H727-R, while metformin was still able to decrease mTOR phosphorylation (−66% ± 21%, *p* < 0.05 vs. basal and −41 ± 10%, *p* < 0.05 vs. basal, respectively) (Figure 7A,B).

Finally, we analyzed the effects of metformin and everolimus on Akt phosphorylation in QGP-1-R and H727-R. As shown in Figure 7C,D, everolimus increased Akt phosphorylation in both resistant cell lines (+116% ± 45%, *p* < 0.01 vs. basal in QGP-1-R, +50% ± 1%, *p* < 0.01 vs. basal in H727-R), suggesting the involvement of the Akt pathway in everolimus resistance. Metformin did not affect Akt phosphorylation in QGP-1-R cells, though it significantly decreased Akt phosphorylation in H727-R cells, confirming the data shown in sensitive QGP-1 and H727 cells.

## 3. Discussion

Currently available treatment options for advanced NETs are rarely curative. Consequently, new treatment strategies are needed. In particular, the inhibitor of the mammalian target of rapamycin (mTOR), everolimus, is recommended by the European Neuroendocrine Tumor Society (ENETS) guidelines for progressive, G1/G2 PAN-NETs [24]. Unfortunately, the development of drug resistance limits its efficacy [8]. The mechanisms leading to resistance are not well understood, but the activation of the Akt pathway seems to be involved [11]. Considering that the most potent anticancer effect of metformin is the suppression of mTOR pathway, a key player in the regulation of cell-growth, metabolism and survival [25], we investigated the effects of the metformin and everolimus combination. According to several studies showing an anticancer effect of metformin in vitro and since the complete medium of NET cells contains high amounts of growth factors and glucose that could affect metformin action [26], we first tested the effect of metformin at millimolar ranges of concentration.

We first focused on primary NET cells, but, considering their low viability after a prolonged time of treatment with metformin, we used it for a relatively short incubation time and at high concentration (10 mM) in combination with everolimus at 10 nM. We demonstrated that metformin and everolimus alone showed antiproliferative effects on both PAN-NET and PNT primary cells. The two compounds in combination were more effective than each monotherapy only on PAN-NET cell proliferation, whereas no additive effects on PNT cells were shown. We confirmed the results obtained on primary cell proliferation in the corresponding cell lines, QGP-1 and H727.

We speculated that the combination might need a prolonged time of incubation to exert a more efficacious antineoplastic effect on pulmonary NET cells. In this respect, a recent study on human glioblastoma stem cells demonstrated that the efficacy of metformin as an anticancer agent is not only dependent on its concentration, but also on the duration of treatment [27]. Indeed, we recently demonstrated that metformin has dose- and time-dependent effects on QGP-1 cells. In particular, metformin at 100 μM and 400 μM was significantly able to exert antiproliferative effects on QGP-1 cell lines at a prolonged time of incubation [22]. In this respect, Vacante et al. considered metformin 400 μM to be comparable to the human therapeutic dose in hepatocellular carcinoma cells [28]. Interestingly, after a prolonged time of incubation, metformin and everolimus in combination were more effective than each agent alone in inhibiting cell viability and the ability of both QGP-1 and H727 cells to form colonies, a crucial event involved in tumorigenesis.

Since NET 3D spheroids mimic the in vivo NET xenograft tumor geometrically and molecularly when compared to monolayer 2D cells [29], and 3D cell culture is becoming an efficient method for drug screening [30], we investigated the impact of metformin and everolimus, alone and in combination, on the growth of QGP-1 and H727 spheroids. Interestingly, metformin and everolimus treatment alone was able to block the growth rate of QGP-1 and H727 3D spheroids, with the combination becoming even more potent in abolishing the spheroids’ growth.

These results confirm that combinatory treatment may actually display greater efficacy than the single monotherapies in both pancreatic and pulmonary NET models and support a potential clinical application of the metformin and everolimus combination. In particular, given the indolent behavior that characterizes most well-differentiated neuroendocrine tumors, chronic treatment with everolimus in combination with metformin at therapeutic doses is likely to provide benefits in terms of tumor growth, without significant additional toxic effects, as metformin is generally well-tolerated [28].

We examined whether the antineoplastic effect of metformin and everolimus is mediated by the induction of apoptosis. Metformin induced a significant increase in QGP-1 and H727 cell apoptosis, as already demonstrated in pancreatic cancer [31] and melanoma cells [32], but no effect on apoptosis was observed when NET cells were treated with everolimus.

We speculated that the combination of metformin and everolimus may induce a more potent mTOR blockade, potentially counteracting the resistance mechanisms induced by everolimus. Consistently with these observations, metformin and everolimus decreased mTOR phosphorylation in QGP-1 and H727 cells and their combination was more effective than monotherapy.

Overactivation of PI3K/Akt has been documented in multiple cancer types, including different NETs [33,34,35]. In fact, Akt plays a key role in many cellular processes [36], such as apoptosis, cell migration and proliferation, as well as glucose metabolism. Therefore, to deepen the understanding of the molecular mechanism involved in the anticancer action of metformin and everolimus, we evaluated Akt phosphorylation. We demonstrated that in QGP-1 cells metformin did not affect Akt phosphorylation, suggesting the involvement of a different pathway in mediating its effects. In this respect, it is noteworthy that metformin acts via AIP to inhibit mTOR phosphorylation in QGP-1 cells [22]. We speculated that the more potent effect of the combination of metformin plus everolimus in inhibiting cell proliferation, colony formation and mTOR phosphorylation in QGP-1 cell lines was due to the action of the AIP pathway, which is involved in cell growth [22].

On the contrary, we demonstrated that metformin significantly decreased Akt phosphorylation in H727 cells and it was also able to suppress Akt phosphorylation when combined with everolimus.

To clarify the role of metformin in the context of everolimus resistance and to confirm the mechanism involved in the anticancer action of the metformin and everolimus combination, we developed everolimus-resistant QGP-1-R and H727-R cells. As expected, everolimus did not affect cell proliferation in QGP-1-R and H727-R, whereas metformin kept inhibiting cell proliferation and inducing apoptosis in both everolimus-resistant cell lines. Importantly, metformin was still able to decrease mTOR phosphorylation in both QGP-1-R and H727-R. These data suggest the use of metformin as a potential therapeutic option for NET patients resistant to everolimus treatment.

Thereafter, we showed the involvement of p-Akt in the mechanism of resistance induced by everolimus, which significantly increased Akt phosphorylation in QGP-1-R and H727-R cells. This aspect is in line with the hypothesis that mTOR inhibition by everolimus induces a feedback loop, leading to Akt activation and consequent mTOR reactivation [37].

Metformin did not affect Akt phosphorylation in QGP-1-R cells, though it was able to reverse everolimus-induced Akt phosphorylation in H727-R cells. These data confirm the involvement of two different pathways leading to mTOR suppression.

In line with what was shown in breast cancer, in which metformin and everolimus have synergistic anticancer effects [38], this study demonstrated that the combination therapy is more effective than the single monotherapies on the inhibition of pancreatic NET cell proliferation, and in counteracting the formation of colonies and inhibiting mTOR phosphorylation in both pancreatic and pulmonary NET cells. Interestingly, the combination of the two compounds displays augmented anticancer activity through the involvement of two different pathways: Akt-independent in the QGP-1 cell line (Figure 8A) and Akt-dependent in the H727 cell line (Figure 8B).

We recently demonstrated that AIP silencing abrogated metformin’s modulation of AIP partners’ expression—AHR, Zac1 and HSP70—providing a plausible mechanistic role of AIP as a regulator of the AIP-AHR-Zac1-HSP70 complex in PAN-NETs [22]. Interestingly, the transcription factor ZAC1 is one of the targets of glycogen synthase kinase 3b (GSK3β) [39]. This kinase regulates the expression of genes involved in cell cycle control and, as reported by Prada et al., its over-activation, combined with decreased baseline IRS-1 protein levels, may be a crucial feature of everolimus resistance [40]. Moreover, previous studies demonstrated that the anticancer action of metformin is mediated by the GSK3 protein [16]. Thus, we can speculate the potential involvement of GSK3 in mediating the Akt-independent mechanism of everolimus-induced resistance shown in QGP-1 cells.

Because of the promising effects of the everolimus and metformin combination in NET cells and spheroids, our results provide a rationale for its use in NET patients. Moreover, the efficacy of metformin as an antineoplastic agent in everolimus-resistant NET cells suggests its promising clinical application as an alternative to everolimus in patients who show resistance to everolimus treatment.

## 4. Materials and Methods

### 4.1. Neuroendocrine Tumor Cell Cultures

The study was approved by the Independent Ethics Committee of Istituto Clinico Humanitas (IRCCS), Rozzano (Milan, Italy), and conformed to the ethical guidelines of the Declaration of Helsinki (no. 1937, 6 February, 2018). Informed consent was obtained from all subjects involved in the study. Human neuroendocrine cells were obtained from 3 PNTs and 3 PAN-NETs (Appendix A), And characterized as previously reported [20]. Briefly, NET tissues were enzymatically dissociated in DMEM containing 2 mg/mL collagenase (Sigma–Aldrich Corporate Headquarters, St. Louis, MO, USA) at 37 °C for 2 h [41,42]. After that, a cell strainer was used, with a strong nylon mesh with 100 µm micron pores designed for isolating primary cells to consistently obtain a uniform single cell suspension from tissues. Dispersed cells were cultured in DMEM supplemented with 10% fetal bovine serum, 2 mM glutamine and antibiotics. Human pancreatic endocrine QGP-1 cell line (Carcinoembryonic antigen (CEA)-producing human pancreatic islet cell carcinoma, JCRB0183, Japanese Homo Sapiens) and H727 cell line (human typical bronchial carcinoid cell line) were grown in RPMI 1640 medium supplemented with 10% FBS, 2 mM glutamine and antibiotics. Cell lines were cultured at 37 °C in 5% CO_2_ atmosphere. QGP-1 and H727 cell lines were authenticated by BMR Genomics srl (Padova, Italy), according to Cell ID TM System (Promega, Madison, WI, USA) protocol and using Genemapper ID Ver 3.2.1, to identify DNA Short Tandem Repeat (STR) profiles.

### 4.2. Development of Everolimus-Resistant QGP-1 and H727 Cell Lines

QGP-1 and H727 cell lines were treated for 24 weeks with 10 nM everolimus. Medium supplemented with 10 nM everolimus was changed every 3 days. Two independent everolimus-resistant QGP-1 and H727 cell lines were established (R and R2). All experiments shown were performed using QGP-1-R and H727-R; R2 cell lines were used to confirm the results. No morphological changes were seen between QGP-1 and H727 cells and QGP-1R and H727-R cell lines.

### 4.3. Proliferation Assay

Cell proliferation was assessed by colorimetric measurement of 5-bromo-2′-deoxyuridine (BrdU) incorporation during DNA synthesis in proliferating cells, according to the manufacturer’s instructions, as previously reported [43]. Briefly, after 24 h serum starvation, cells were incubated with metformin (1 mM, 10 mM) (#1396309, Sigma Aldrich Corporate Headquarters, St. Louis, MO, USA), or treated with everolimus (10 nM, 100 nM), or with the combination of everolimus 10 nM or 100 nM and metformin 1 mM or 10 mM for 24 h. Afterwards, the cells were incubated with BrdU for 2 h (cell lines) and 24 h (primary cultures) to allow BrdU incorporation in newly synthesized cellular DNA. All experiments were repeated at least three times and each determination was done in triplicate.

### 4.4. Apoptosis Analysis by Flow Cytometry

Detection of apoptosis was performed using the Annexin V/PI kit (#640932, Biolegend, San Diego, CA, USA), according to the manufacturer’s protocols. After 48 h of metformin incubation (10 mM) or everolimus (10 nM), alone or in combination, cells were double-stained with fluorochrome-labeled annexin V and red-fluorescent propidium iodide dye. The percentage of apoptotic cells was determined by flow cytometry using Fluorescence-activated cell sorting _(FACS)LSRFortessa (BD Biosciences, Franklin Lakes, NJ, USA). The results were analyzed using FACSDiva software (BD Biosciences).

### 4.5. Cell Viability Assay

Cell viability was assessed in QGP-1 and H727 cells by 3-(4,5-dimethylthiazol-2-yl)-2,5-diphenyltetrazolium bromide (MTT) (Sigma-Aldrich, St. Louis, MO, USA) conversion into formazan by mitochondrial enzymes, thus determining the number of metabolically active cells. Cells were seeded at a cell density of 5 × 10^4^ cells/well in 96-well plates at 37 °C. Cells were incubated with metformin (50 µM, 100 µM and 400 µM), or treated with everolimus (10 nM), or with the combination of everolimus 10 nM and metformin (50 µM, 100 µM and 400 µM) for 7 days. MTT was dissolved in sterile phosphate-buffered saline (PBS) at 5 mg/mL and 10 µL of the obtained solution was added to each well for a final concentration of 0.5 mg/mL, followed by incubation at 37 °C for 4 h to form purple-colored formazan crystals. Subsequently, the MTT solution was removed and the cells/crystals were solubilized by adding 10% SDS solution in 0.01 M HCl overnight in a humidified 5% CO_2_ incubator. Absorbance was read at 570 nm in a 96-well plate in a Synergy H4 plate reader. Each determination was done in triplicate. Experiments were repeated at least three times.

### 4.6. Colony Formation

QGP-1 or H727 cells were seeded into 24-well plates and 96-well plates in triplicate at a density of 1250 cells or 500 cells, respectively. Then, cultured cells were incubated at 37 °C for 7 days with metformin (100 μM and 400 μM) and everolimus 10 nM, alone or in combination. To count the number of cells, cells were incubated with the CyQUANT^®^ GR dye/cell-lysis buffer, according to the manufacturer’s protocols (#C7026, Invitrogen, Waltham, MA, USA). Number of cells were then quantified by using a fluorescence microplate reader. Images of QGP-1 and H727 clones were acquired using brightfield microscopy with a 4X objective lens (Widefield IX53 Inverted, Olympus, Tokyo, Japan). Average colony area was measured using the NIH software ImageJ.

### 4.7. 3D Cultures (Spheroids) and Growth Measurement

For sphere formation, a suspension of QGP-1 or H727 cells was inoculated in Biomimesys^®^ (Celenys, Rouen, France), following the manufacturer’s instructions. Briefly, 50,000 QGP-1 or H727 cells were plated in 3D culture—96-well plates containing a hyaluronic acid matrix. When the 3D spheroids reached around 100–200 μm in diameter at day 4, they were treated with metformin (400 μM) and everolimus 10 nM, alone or in combination. This is denoted as day 0 for drug treatment.

Photographs of spheroids were acquired at days 0, 3 and 6, using brightfield microscopy with a 4X objective lens (Widefield IX53 Inverted, Olympus). Images were calibrated, using µm as the unit for calculation. The area of the spheroids was determined using the ImageJ software and values are shown as a percentage of control at day 0, as recently reported [44]. Experiments were done in triplicate and repeated two times.

### 4.8. Cell Cycle Analysis

Cells were cultured in 6-well plates (6 × 10^5^ cells per well) with 1 mL of medium. After 24 h of metformin 10 mM and everolimus 10 nM incubation, QGP-1-R and H727-R cells were harvested and washed twice in PBS 1× at 4 °C. Then cells were fixed in 1mL of cold 80% ethanol added dropwise to the pellet while vortexing and incubated at 4 °C for 30 min. After that, the cells were centrifuged at 2000 rpm for 5 min to remove ethanol and then washed twice in PBS 1× at 4° C. Finally, the cells were stained for 30 min in DAPI solution (PBS, 0.01% TRITON X-100, 1 ug/mL DAPI-Invitrogen™) at room temperature (RT) and then DNA content was read using BD FACSymphony A5. The results were analyzed using FlowJo^®^ software (FlowJo LLC, Ashland, OR, USA).

### 4.9. Immunoblot Detection of Phospho-mTOR and Phospho-Akt

To detect mTOR and Akt phosphorylation, QGP-1 and H727 cells were treated for 24 h with or without 10 mM metformin and 10 nM everolimus, alone or in combination. Immunoblotting analysis was performed with 1:1000 dilution of phospho-mTOR antibody and 1:1000 dilution of phospho-Akt ser473 antibody (#2971, Cell Signaling Technology, Danvers, MA, #MAB-94111 Immunological Sciences, Rome, Italy). Monoclonal anti-GAPDH (AM4300, Ambion, 1:4000) or anti-vinculin (1:1000 dilution, Cell Signaling Technology, Danvers, MA, USA) were used as housekeeping genes, developed using an anti-mouse or anti rabbit HRP-linked antibody (#7076, Cell Signaling Technology 1:2000 and #7074, Cell Signaling Technology, 1:2000, respectively).

### 4.10. Statistical Analysis

The results were expressed as the mean ± SD. A paired two-tailed Student’s test was used to detect the significance between two series of data.

One-way ANOVA analysis was used to compare three or more groups, followed by Tukey’s multiple comparisons test. Dunnett’s method was used to test two or more experimental groups against a single control group. For spheroid size analysis, a two-way ANOVA followed by Tukey’s multiple comparisons test was used. Calculations were performed using GraphPad Prism 8.0 software (GraphPad Software, Inc., La Jolla, CA, USA). *p* < 0.05 was accepted as statistically significant.

## 5. Conclusions

Considering the promising effects of the everolimus and metformin combination in NET cells and spheroids, our results provide a rationale for its use in NET patients. In addition, the antiproliferative and proapoptotic effect of metformin in everolimus-resistant NET cells suggests its promising clinical application in patients resistant to everolimus treatment.

## Figures and Tables

**Figure 1 cancers-12-02143-f001:**
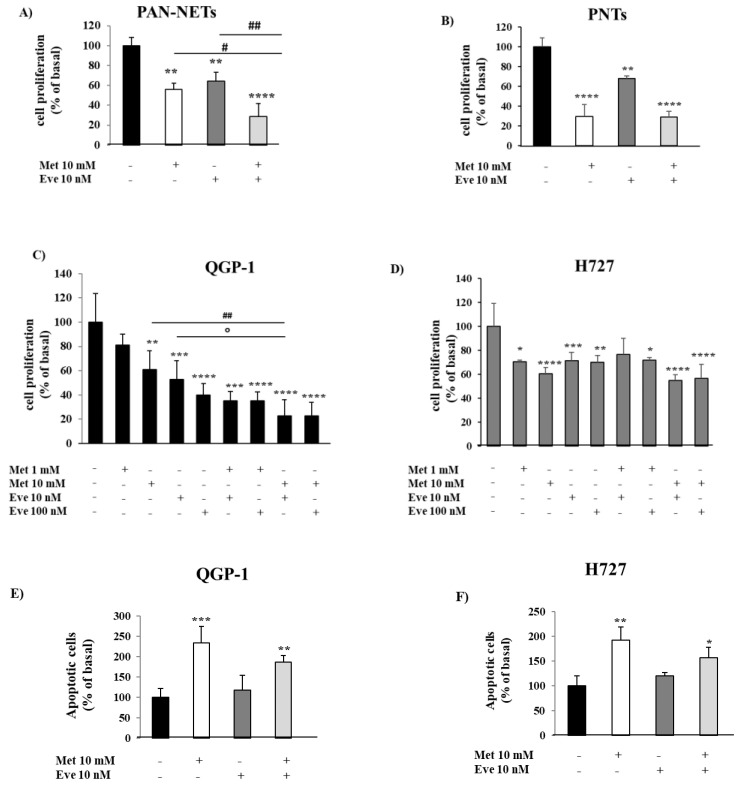
The effect of metformin (Met) and everolimus (Eve) on neuroendocrine tumor (NET) cell proliferation and apoptosis. (**A**,**B**) To measure cell proliferation, we incubated primary NET cells with metformin 10 mM and everolimus 10 nM, alone or in combination, for 24 h, then with BrdU for 24 h. Experiments were repeated 3 times and each determination was done in triplicate. Basal = untreated control. Values represent mean (± SD.) ** = *p* < 0.01, **** = *p* < 0.0001 vs. corresponding basal, # = *p* < 0.05, ## = *p* < 0.01 vs. single drug. Statistical analysis was performed with a one-way ANOVA followed by Tukey’s post-hoc test. (**C**,**D**) To measure cell proliferation, QGP-1 and H727 cell lines were incubated for 24 h with metformin (1–10 mM) and everolimus (10–100 nM), alone or in combination, and then treated with BrdU for 2 h. Experiments were repeated at least 4 times and each determination was done in triplicate. Values represent mean (±SD.) * = *p* < 0.05, ** = *p* < 0.01, *** = *p* < 0.001, **** = *p* < 0.0001 vs. corresponding basal, ## = *p* < 0.01 vs. metformin, ° = *p* < 0.05 vs. everolimus. (**E**,**F**) The graph shows the percentage of apoptotic cells after metformin (10 mM) and everolimus (10 nM) treatment, alone or in combination, compared to basal. Only metformin 10 mM significantly increased the rate of apoptosis. Values represent mean ± SD of 3 experiments. * = *p* < 0.05, ** = *p* < 0.01, *** = *p* < 0.001 vs. corresponding basal. Statistical analysis was performed with a one-way ANOVA test followed by Dunnett’s post-hoc test.

**Figure 2 cancers-12-02143-f002:**
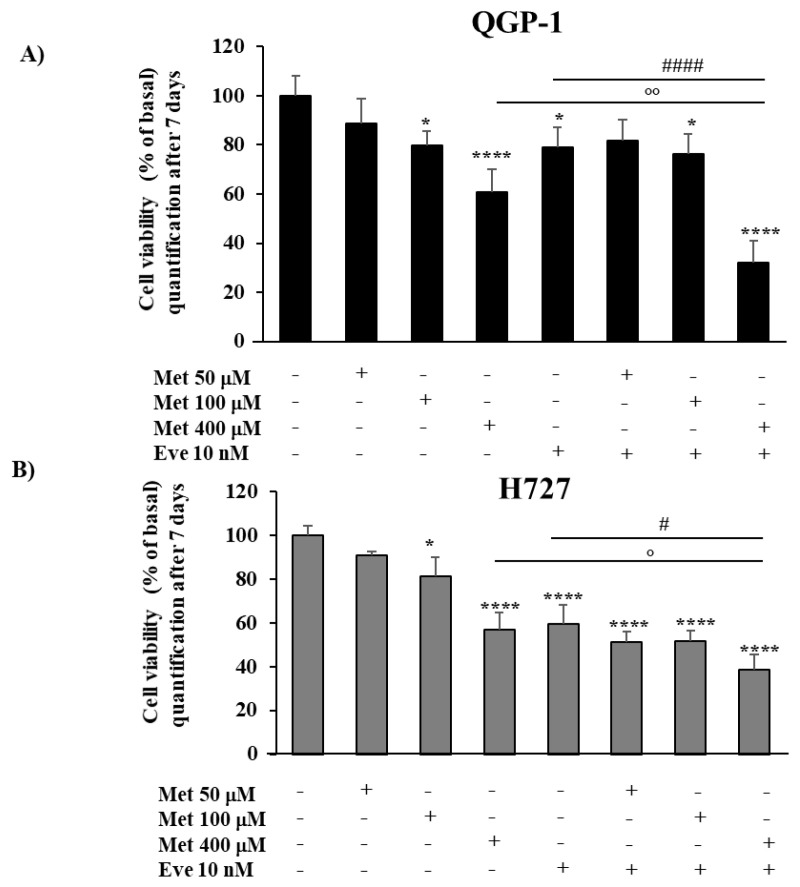
The effect of metformin (Met) and everolimus (Eve) on QGP-1 and H727 cell viability. (**A**,**B**) Cells were incubated with metformin (50 µM, 100 µM and 400 µM), or treated with everolimus (10 nM), or with the combination of everolimus 10 nM and metformin (50 µM, 100 µM and 400 µM) for 7 days. The combination of metformin 400 µM and everolimus 10 nM induced a more effective reduction of both cell lines’ viability than each monotherapy. Values represent mean (± SD.) * = *p* < 0.05, **** = *p* < 0.0001 vs. corresponding basal, # = *p* < 0.05 and #### = *p* < 0.0001 vs. everolimus, ° = *p* < 0.05 and °° = *p* < 0.01 vs. metformin. Experiments were repeated at least 3 times and each determination was done in triplicate. Statistical analysis was performed with a one-way ANOVA test followed by Tukey’s post-hoc test.

**Figure 3 cancers-12-02143-f003:**
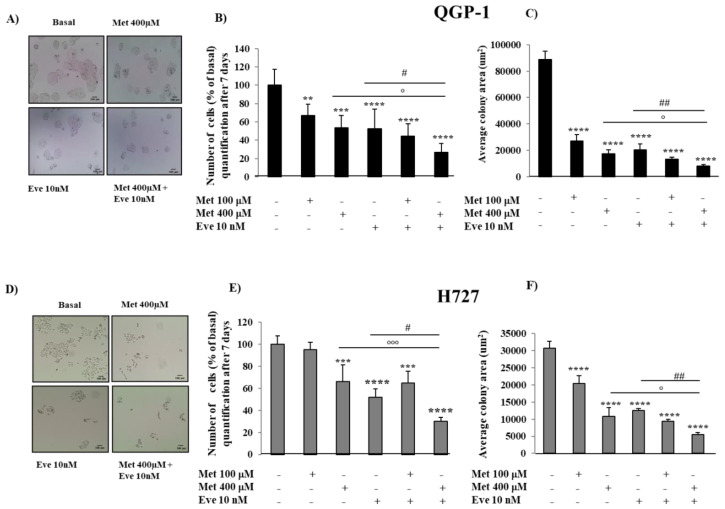
The effect of metformin (Met) and everolimus (Eve) on QGP-1 and H727 colony formation. (**A**,**D**) QGP-1 and H727 cells incubated with or without metformin 400 µM, everolimus 10 nM or with their combination. Microscopy images show that the combination of metformin and everolimus strongly decreased QGP-1 and H727 cell number and colony size. Scale bar 100 μm**.** (**B**,**E**) To count the number of cells, after metformin treatment (100 µM and 400 µM) and everolimus (10 nM), alone or in combination, for 7 days, QGP-1 and H727 cells were incubated with the CyQUANT^®^ GR dye and fluorescence was measured at 480/520 nm. The graph shows that the metformin and everolimus combination is more effective than each monotherapy in decreasing the number of QGP-1 and H727 cells. Values represent mean ± SD of at least 3 experiments. ** = *p* < 0.01, *** = *p* < 0.001 **** = *p* < 0.0001 vs. corresponding basal, ° = *p* < 0.05 and °°° = *p* < 0.001 the combination vs. metformin, # = *p* < 0.05 the combination vs. everolimus. Statistical analysis was performed with a one-way ANOVA test followed by Tukey’s post-hoc test. (**C**,**F**) To measure colony surface area, 3–4 fields were randomly selected in each well. Average of colony area (µm^2^) was measured using the software ImageJ. Experiments were repeated at least 3 times. Values represent mean (±SD.) **** = *p* < 0.0001 vs. corresponding basal, ° = *p* < 0.05 vs. metformin, ## = *p* < 0.01 vs. everolimus. Statistical analysis was performed with a one-way ANOVA test followed by Tukey’s post-hoc test. Scale bar of (**A**) Scale bar 100 μm.

**Figure 4 cancers-12-02143-f004:**
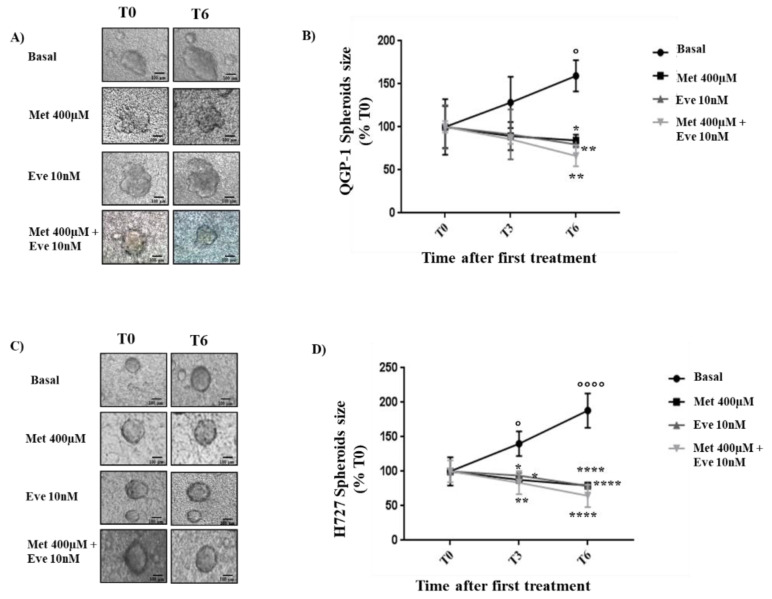
Metformin (Met) and everolimus (Eve) treatment on QGP-1 and H727 3D Spheroids. QGP-1 And H727 spheroids were performed using the BIOMIMESYS^®^ hydroscaffold of crosslinked hyaluronic acid (HA) chains. Cells were followed by imaging using brightfield microscopy with a 4X objective lens (Widefield IX53 Inverted, Olympus). (**A**,**C**) Representative images of QGP-1 and H727 spheroids after their formation and after three and six days of treatment with metformin 400 µM and everolimus (10nM), alone or in combination. 0 days: time point when treatments started (T0); 3 days and 6 days are the time points analyzed after the treatments started (T3–T6). Scale bar 100 µm. (**B**,**D**) Growth of the 3D spheroids upon treatment with metformin and everolimus, alone or in combination**.** The area was measured using the software ImageJ. Experiments were repeated 2 times and each determination done in triplicate. Statistical analysis was performed with a two-way ANOVA test followed by Tukey’s post-hoc test. Values represent mean (±SD.) and are shown as a percentage of control at day 0. * = *p* < 0.05, ** = *p* <0.01, **** = *p* < 0.0001 vs. T6 Basal. ° = *p* < 0.05 vs. T0 Basal, °°°° = *p* < 0.0001 vs. T0 Basal.

**Figure 5 cancers-12-02143-f005:**
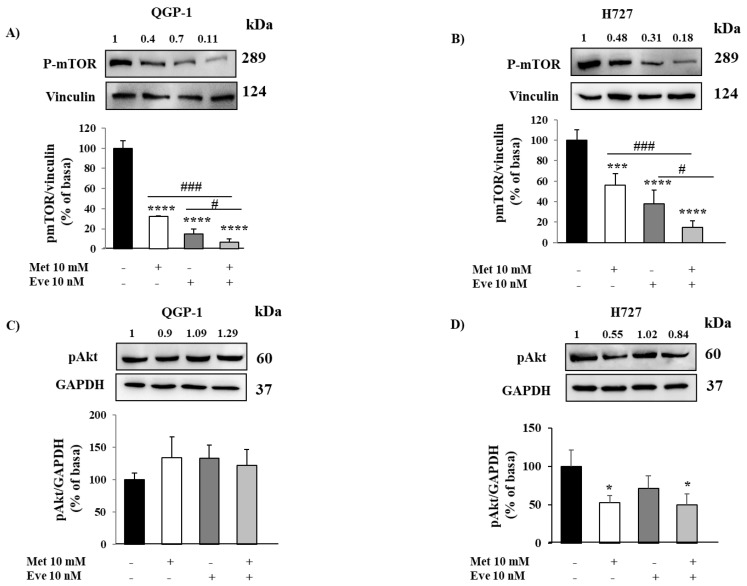
The effect of metformin (Met) and everolimus (Eve) on mTOR and Akt phosphorylation in QGP-1 and H727 cells. (**A**,**B**) Representative immunoblots of phospho-mTOR demonstrating that the inhibition of mTOR by metformin and everolimus in combination is more potent than each monotherapy in both cell lines. The graph shows the quantification of phospho-mTOR normalized to the housekeeping gene vinculin (mean value ± SD from 3 independent experiments). *** = *p* < 0.001, **** = *p* < 0.0001 vs. corresponding basal and # = *p* < 0.05, ### = *p* < 0.001 vs. the single drug. Statistical analysis was performed with a one-way ANOVA test followed by Tukey’s post-hoc test. (**C**) Representative immunoblots of phospho-Akt demonstrating that metformin does not affect Akt phosphorylation in the QGP-1 cell line. The graph shows the quantification of phospho-Akt normalized to the housekeeping gene GAPDH (mean value ± SD from 3 independent experiments). Statistical analysis was performed with a one-way ANOVA test followed by Tukey’s post-hoc test. (**D**) Representative immunoblots of phospho-Akt demonstrating that metformin significantly decreases Akt phosphorylation in H727 cell lines. The graph shows the quantification of phospho-Akt normalized to the housekeeping gene GAPDH (mean value ± SD from 3 independent experiments). * = *p* < 0.05 vs. corresponding basal. Statistical analysis was performed with a one-way ANOVA test followed by Dunnett’s post-hoc test. Unprocessed images for WB results see Appendix A.

**Figure 6 cancers-12-02143-f006:**
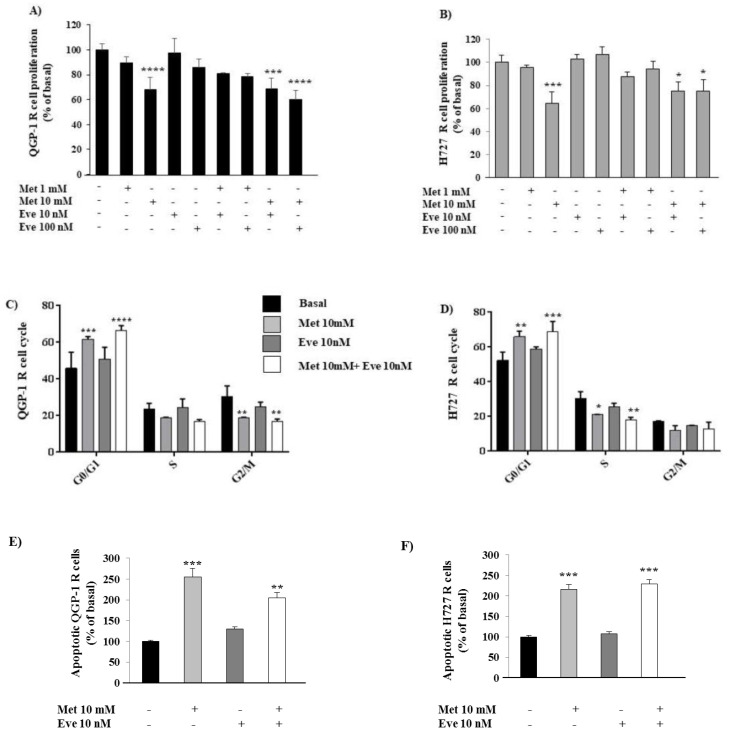
Metformin (Met) continued to exert its anticancer effects on cell proliferation, cell cycle and apoptosis in QGP-1-R and H727-R cells. (**A**,**B**) To measure cell proliferation, everolimus-resistant QGP-1-R and H727R cell were treated with metformin (1–10 mM) and everolimus (Eve 10–100 nM) or their combination for 24 h and then with BrdU for 2 h. Experiments were repeated 3 times and each determination was done in triplicate. Values represent mean (± SD.) * = *p* < 0.05, *** = *p* < 0.001, **** = *p* < 0.0001 vs. corresponding basal. (**C**,**D**) QGP-1-R and H727-R cells were treated with metformin (10 mM) and everolimus (10 nM) or their combination for 24 h and their cell cycle was analyzed by flow cytometry. Experiments were repeated at least 3 times. Values represent mean (± SD.) * = *p* < 0.05, ** = *p* < 0.01, *** = *p* < 0.001, **** = *p* < 0.0001 vs. corresponding basal. Statistical analysis was performed with a one-way ANOVA test followed by Dunnett’s post-hoc test. (**E**,**F**) The graph showed the percentage of apoptotic cells after metformin (10 mM) and everolimus (10 nM) treatment, alone or in combination, compared to basal in QGP-1-R and H727-R cells. Only metformin 10 mM significantly promoted apoptosis. Values represent mean ± SD of 3 experiments. ** = *p* < 0.01, *** = *p* < 0.001 vs. corresponding basal. Statistical analysis was performed with a one-way ANOVA test followed by Dunnett’s post-hoc test.

**Figure 7 cancers-12-02143-f007:**
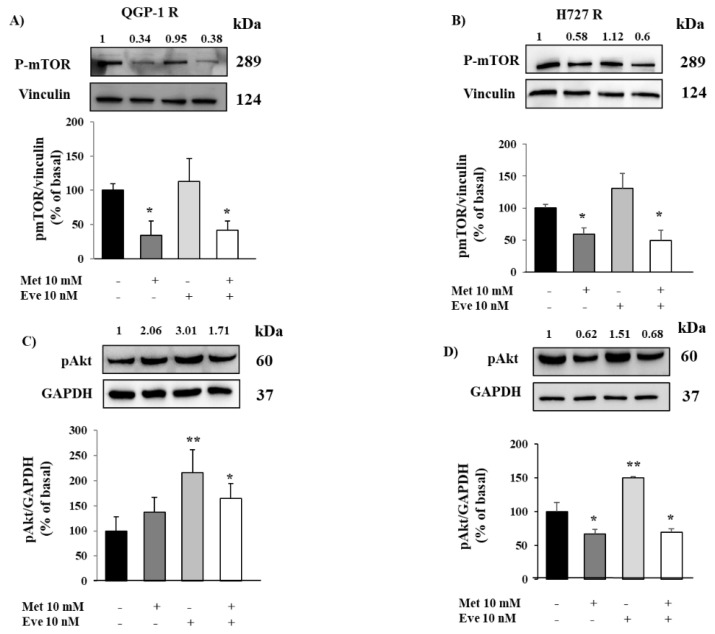
The effect of metformin (Met) on mTOR and Akt phosphorylation in QGP-1-R and H727-R cells. (**A**,**B**) Representative immunoblots of phospho-mTOR demonstrating that metformin continued to inhibit mTOR phosphorylation in everolimus-resistant QGP-1-R and H727-R cells. Graphs show the quantification of phospho-mTOR normalized to the housekeeping gene vinculin (mean value ± SD from 3 independent experiments). * = *p* < 0.05 vs. corresponding basal. Statistical analysis was performed with a one-way ANOVA test followed by Dunnett’s post-hoc test. (**C**,**D**) Representative immunoblots of p-Akt, after metformin and everolimus incubation, alone or in combination. Metformin did not affect Akt phosphorylation in QGP-1-R cells, conversely, significantly decreased p-Akt in H727-R cells. Graphs show the ratio of p-Akt/GAPDH normalized to basal (mean value ± SD from 3 independent experiments). * = *p* < 0.05, ** = *p* < 0.01 vs. corresponding basal. Statistical analysis was performed with a one-way ANOVA test followed by Dunnett’s post-hoc test. Unprocessed images for WB results see Appendix A.

**Figure 8 cancers-12-02143-f008:**
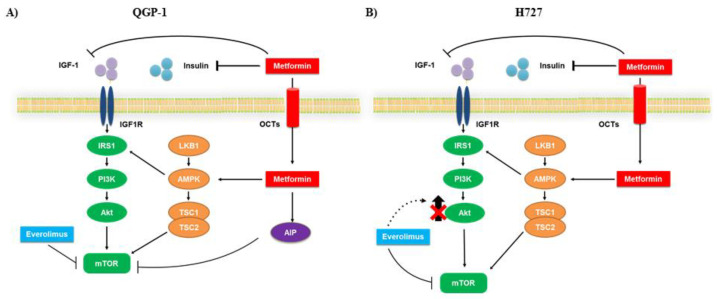
Schematic representation of the possible mechanism of action of metformin and everolimus in QGP-1 and H727 cells. Metformin acts via 5′-AMP-activated protein kinase (AMPK) dependent or independent pathways, leading to mTORC1 complex inhibition. Everolimus inhibits mTOR phosphorylation, but its long-term use is limited by the development of resistance, probably due to the activation of Akt signaling. The combination of metformin and everolimus induces a more potent blockade of mTOR and potentially counteracts the resistance mechanisms triggered by everolimus. (**A**) In QGP-1 cells metformin did not affect Akt phosphorylation, probably acting through the AIP pathway to decrease mTOR phosphorylation. (**B**) In H727 cells, metformin decreases Akt phosphorylation to suppress mTOR phosphorylation.

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
