# Peer review of "Metformin and Everolimus: A Promising Combination for Neuroendocrine Tumors Treatment"

_cancers, 2020, doi:10.3390/cancers12082143_

Round 1
Reviewer 1 Report
The paper might be of interest for readers after the answers to the following comments:
- Indicate the number and data of the Independent Ethics Committee approval of the study, and if it follows the Helsinki guidelines.
- Indicate the stage at diagnoses, treatments (previous/surgical/after diagnoses) as well as the location of each tumor sample for the cultured cells obtained from the surgically removed human NETs.
- Clarify/state the IHQ neuroendocrine features of the cultured cells obtained from the surgically removed human NETs.
- Clarify/state the effects on cell proliferation and apoptosis in each set of cultured cells obtained from the surgically removed human NETs, namely for the PAN-NET G1, G2, and G3 (to be included in a Suppl. File).
- The authors state in the discussion that this is an in vitro. Thus, the authors should refrain the wording “anticancer/anti-tumor) concerning the two drugs (everolimus and metformin) effects in the cultured tumor in vitro cells.
- Clarify/state whether or not the IC50 estimation (e.g., using PrestoBlue assay) was performed concerning the two drugs tested in this study.
- Clarify/state the meaning of “basal”, as stated in various evaluations of the study. Also, if and when applicable, untreated cultured cells were or not appropriately used as controls.
- Clarify/state the meaning of the columns "colors" in Fig 1, 2, and 3.
- Clarify/state in Fig 1 (E and D) which graph indicates QGP-1 and H727. Add results from mono and co-incubation drug effects in Figs whenever tested.
- Include in pictures of Fig 3 (A and D) and Fig 4 (A and C) the same appropriate scale.
- Can this interesting in vitro study be upgraded with the direct evaluation of IRS-1 and AIP expression in the tested cultured cells, to eventually clarify their putative role in neuroendocrine neoplasms?
Reviewer 2 Report
Vitali et al looked at the promising combination of Metformin and Everolimus for pancreatic and pulmonary neuroendocrine tumour treatment. For this purpose they used 2 NET cell lines (QGP-1 and H727) and also cultured cells from 6 patients with neuroendocrine tumours. The authors looked at NET cell proliferation, apoptosis, colony formation, cell viability, NET spheroids growth, Akt and mTOR pathways.
The techniques used are sound and the text flow is good. Justified conclusions at the end.
Couple of questions for clarification:
1. Why did you decide to proceed with different concentrations for the cultured patient cells and the QCP-1 and H727 lines? What would be the effect of metformin 400uM on proliferation, for instance?
2. What criteria did you use when deciding Tukey's vs. Dunnett's test?
PS
It should be noted that it is Tukey's test for post hoc analysis and not Turkey's.
Reviewer 3 Report
Good research for pan-NET, some minor issues: 1) The main part of the assays are mostly based on the cell line work, which generates several problems: a) does the concentration used in the assays applicable in the in vivo environment? b) how specific does the drug work on the target, does it influence other tissue/cell type as well? c) the dosage window safety and efficacy issue. d) are there any reported biological process that could convert the drug lead's active form to render it less effective? It would be preferred if the authors could provide more such background information. 2) For the colony formation/cell persistent growing assays, did the authors check the actual concentration during the time course? especially for the 3D one. I was wondering how effective for them to get inside. This will has an impact to do with the drug administration for in vivo models and future clinical studies. 3) Nowadays, almost anything can fall on the key pivotal axis of mTOR and certainly kinase systems. It would however, extremely inconvenient to regulate the process or related function for these molecules since they tend to interfere other irrelevant physiological events. Have the authors taken any unrelated controls from mTOR/Akt to evaluate the influence for the drug combinations used in this study?
Round 2
Reviewer 1 Report
After the revision by the authors, the paper seems useful to readers.